# Rotavirus and bacterial diarrhoea among children in Ile-Ife, Nigeria: Burden, risk factors and seasonality

**Temiloluwa Ifeoluwa Omotade**[1], **Toluwani Ebun Babalola**[2], **Chineme Henry Anyabolu**[2], **Margaret Oluwatoyin Japhet**[1]*

1 Faculty of Science, Department of Microbiology, Obafemi Awolowo University, Ile-Ife, Osun State, Nigeria,
2 Department of Paediatrics, Obafemi Awolowo University Teaching Hospital (OAUTHC), Ile-Ife, Osun State, Nigeria

* mojaphet@oauife.edu.ng

**Data Availability Statement:** All relevant data are within the manuscript and its Supporting Information files.

## Abstract

### Background

Diarrhoea is a leading cause of death among under-five children globally, with sub-Saharan Africa alone accounting for 1/3 episodes yearly. Viruses, bacteria and parasites may cause diarrhoea. Rotavirus is the most common viral aetiology of diarrhoea in children less than five years globally. In Nigeria, there is scarce data on the prevalence/importance, burden, clinical/risk factors and seasonality of rotavirus and bacteria and this study aims to determine the role of rotavirus and bacteria on diarrhoea cases in children less than five years in Ile-Ife, Nigeria.

### Methods

Socio-demographic data, environmental/risk factors and diarrhoiec stool samples were collected from children less than five years presenting with acute diarrhoea. Rotavirus was identified using ELISA. Bacteria pathogens were detected using cultural technique and typed using PCR. Diarrhoeagenic *E. coli (*DEC) isolates were subjected to antimicrobial susceptibility testing. Pathogen positive and negative samples were compared in terms of gender, age-group, seasonal distribution, and clinical/risk factors using chi-square with two-tailed significance. SPSS version 20.0.1 for Windows was used for statistical analysis.

### Results

At least one pathogen was detected from 63 (60.6%) children having gastroenteritis while 28 (44.4%) had multiple infections. Rotavirus was the most detected pathogen. Prevalence of rotavirus mono-infection was 22%, multiple infection with bacteria was 45%. Mono-infection prevalence of DEC, *Shigella* spp., and *Salmonella* spp. were 5.8% (6/104), 5.8% (6/104), and 2.9% (3/104) and co-infection with RVA were 23.1% (24/104), 21.2% (22/104) and 10.6% (11/104) respectively. All rotaviral infections were observed in the dry season. The pathotypes of *DEC* detected were STEC and EAEC. Parent earnings and mid-upper

**Funding:** The author(s) received no specific funding for this work.

**Competing interests:** The authors have declared that no competing interests exist.

arm circumference measurement have statistical correlation with diarrhoea (p = 0.034; 0.035 respectively).

## Conclusion

In this study, rotavirus was more prevalent than bacteria and occurred only in the dry season. Among bacteria aetiologies, DEC was the most common detected. Differences in seasonal peaks of rotavirus and DEC could be employed in diarrhoea management in Nigeria and other tropical countries to ensure optimal limited resources usage in preventing diarrhoea transmission and reducing indiscriminate use of antibiotics.

## 1. Introduction

The World Health Organization (WHO) reported 1.7 billion cases of childhood diarrhoeal disease globally in 2017, with about 370,000 deaths among children [1,2]. The United Nations International Children's Emergency Fund (UNICEF) in its annual report estimated that diarrhoea kills about 2,195 children daily, globally; this is higher than the combined mortality cases of AIDS, malaria, and measles [3].

One-third of the global annual diarrhoea episodes is associated with Sub-Saharan African countries [4]. The high prevalence of diarrhoea in the region has been attributed largely to the lack of safe water for drinking, indecorous means of human fecal waste disposal, concentrated crowding of basic houses, and poor overall hygiene standards [5]. Given that children's immune system is not fully developed, substandard living conditions increases the vulnerability of children younger than 5 years to diarrhoea when exposed to diarrhoea pathogens [6].

Microorganisms causing diarrhoea are transmitted through contaminated water, and food [7,8]. Three major groups of microorganisms have been implicated as causal agents of diarrhoea. They include viruses (rotaviruses, caliciviruses, astroviruses and adenoviruses), bacteria (Diarrhoegenic *E. coli*, *Shigella species*, *Salmonella species*, *Campylobacter jejuni*), and parasites (*Entamoeba histolytica*, *Giardia lambia*) [9–11]. Viral agents are the major aetiological agents of diarrhoea in Africa [12]. Rotavirus A (RVA) has been reported as the common cause of severe diarrhoea in infants and young children worldwide [1,10,13], associated with >50% of gastroenteritis in this age group [14].

Although there has been a consistent decline in childhood mortality rate caused by diarrhoea over the years, diarrhoea still remains a great burden in Nigeria, putting the country as one of the countries with high under-five diarrhoea mortality rate [15,16]. To reduce diarrhoea related infant morbidity and mortality in countries with high incidence, epidemiological surveys of childhood diarrhoea, in addition to diarrhoea aetiology information is useful in planning, implementation of control strategies and treatment [17]. In Nigeria, there is dearth of single-study data on importance of rotavirus and bacteria as diarrhoea aetiology in children, the associated burden, environmental/risk factors and the seasonality. We therefore investigated the role of rotavirus and bacteria in diarrhoea disease, the associated risk/environmental factors, the seasonal distribution of pathogens as well as the antibiotic sensitivity pattern of the bacteria isolates among children less than 5 years in Ile-Ife, Osun state, south-western Nigeria. This will serve as baseline data for diarrhoea pathogens. It will also inform appropriate policies for diarrhoea prevention and reduce indiscriminate antibiotics use, hence improving diarrhoea management in children.

## 2. Materials and methods

### 2.1 Study design

This prospective observational study was designed for one year sample collection, between April 2019 and March 2020, however, samples could not be collected throughout December and part of March due to Lassa fever outbreak and COVID-19 respectively. Four health centres were used as study sites namely: Children emergency ward and the children outpatient/diarrhoea unit Obafemi Awolowo University Complex Teaching Hospital, Oke-Ogbo state Hospital, Urban Comprehensive Health Center Eleyele and Enuowa primary health Centre Ile-Ife, all in southwest Nigeria. Children less than 5 years with acute diarrhoea episodes (diarrhoea is defined as three or more watery stool over a 24-hour period, with or without other symptoms such as fever, vomiting or dehydration) and whose parents give written informed consent were included in the study. Exclusion criteria were children who do not have diarrhoea, diarrhoeic children older than 5 years old, and those whose parent/guardian do not give formal consent to enroll their children in the study. Sociodemographic data (age, sex, monthly income, family size, etc.), clinical and risk factors (symptoms, source of drinking water, six months exclusive breast feeding, stool description, knowledge about RVA etc.) were collected using structured questionnaire completed by the parent or guardian of the children.

### 2.2 Ethics approval

Ethical approval (ERC/2018/02/14) was obtained from the Obafemi Awolowo University Complex Teaching Hospital (OAUTHC) Research Ethics Committee (OAUTHC REC) of the Obafemi Awolowo University Ile-Ife, Nigeria.

### 2.3 Detection and identification of bacteria

Faecal samples were inoculated onto Eosin Methylene Blue (EMB) agar for isolation of *Escherichia coli* as well as *Salmonella-Shigella* agar (SSA) plates for the isolation of *Salmonella* and *Shigella* species. The plates were incubated for 24 hours aerobically at 37°C. Distinct colonies were picked and identified by standard biochemical tests. Cultures with no growth after 24 hours of incubation were recorded as negative for bacteria growth.

### 2.4 DNA extraction

The DNA of *the E. coli* isolates were extracted using boiling method. Briefly, distinct colonies of each bacteria isolate were suspended in 100μl distilled water in a clean Eppendorf tube until emulsification was achieved; the resulting mixture was boiled at 100℃ for 15 minutes and finally centrifuged at 10,000 revolutions per minutes for five minutes in a microcentrifuge.

The supernatant containing the DNA was separated into a different sterile Eppendorf tube and stored at -20°C till further analysis.

### 2.5 Molecular identification of isolates by amplification of *E. coli* 16S rRNA gene

The isolated organisms suspected to be *E. coli* by their cultural and biochemical characteristics were confirmed by polymerase chain reaction (PCR) using published primers specific to *E. coli* 16S rRNA gene (Table 1), adopting the procedure described by Odetoyin et al. [18].

**Table 1. Polymerase chain reaction primers for diarrhoeagenic *Escherichia coli*.**

| Type | Primer Designation | Primers (5 to 3) | Target gene | Amplicon or probe size (bp) | Reference |
|------|-------------------|------------------|-------------|----------------------------|-----------|
| ECO | ECO-1 | GACCTCGGTTTAGTTCACAGA | 16srRNA | 585 | 19 |
|  | ECO-2 | CACACGCTGACGCTGACCA |  |  | 19 |
| EPEC | eae 1 | CTGAACGGCGATTACGCGAA | Eae | 917 | 19 |
|  | eae 2 | CCAGACGATACGATCCAG |  |  | 19 |
|  | bfp 1 | AATGGGCTTGCGCTTCCAG | bfpA | 326 | 19 |
|  | bfp 2 | GCCGCTTTATCCAACCTGGTA |  |  | 19 |
| EAEC | EAEC1 | CTGGCGAAAGACTGTATCAT | CVD432 | 630 | 19 |
|  | EAEC2 | CAATGTATAGAAATCCGCTGTT |  |  | 19 |
| ETEC | LTf | GGCGACAGATTATACCGTGC | LT | 450 | 19 |
|  | LTr | CAATGTATAGAAATCCGCTGTT |  |  | 19 |
|  | STf | ATTTTTMTTTCTGTATTRTCTT | ST | 190 | 19 |
|  | STr | CACCCGGTACARGCAGGATT |  |  | 19 |
| EIEC | IpaH1 | GTTCCTTGACCGCCTTTCCGATACCGTC | ipaH | 600 | 19 |
|  | IpaH2 | GCCGGTCAGCCACCCTCTGAGAGTAC |  |  | 19 |
| EHEC | Stx1f | ATAAATCGCCATTCGTTGACTAC | Stx1 | 180 | 19 |
|  | Stx1r | AGAACGCCCACTGAGATCATCC |  |  | 19 |
|  | Stx2f | GGCACTGTCTGAAACTGCTCC | Stx2 | 255 | 19 |
|  | Stx2r | TCGCCAGTTATCTGACATTCTG |  |  | 19 |

EPEC- Enteropathogenic *E. coli* EHEC- Enterohemorrhagic *E. coli*.

EAEC-Enteroaggregative *E. coli* ETEC- Enterotoxigenic *E. coli* EIEC–Enteroinvasive *E. coli*.

## 2.6 Screening for diarrhoeagenic *E. coli*

All *E. coli* isolates were screened for virulence genes (Table 1) of five different pathotypes of diarrhoeagenic *E. coli*, including enterotoxigenic *E. coli* (ETEC), enteroinvasive *E. coli* (EIEC), enteropathogenic *E. coli* (EPEC), enteroaggregative *E. coli* (EAEC), and enterohaemorrhagic *E. coli* (EHEC) as previously described by Aranda et al., [19].

## 2.7 Antimicrobial susceptibility testing

Antibiotic susceptibility testing of DEC isolates was done using the disc diffusion method. Commonly used and available antibiotics (Oxoid Ltd, UK) impregnated discs of known concentration including; Cephalosporin [cefoxitin (30μg), cefuroxime sodium (30μg)], Sulfonamide [sulphamethoxazole/Trimethoprim (25μg)], Chloramphenicol [chloramphenicol (30μg)], Fluoroquinolones [Ciprofloxacin (5μg), Ofloxacin (5μg)], Tetracycline [Doxycycline (30μg)], and Carbapenem [imipenem (10μg)] were tested on all the DEC isolate. Antibiotic susceptibility or multidrug resistance (resistance of an organism to 3 or more different classes of antibiotics) in isolates was determined by interpreting the zones of inhibition into susceptible or resistance using the clinical and laboratory standards institute guide [CLSI 2019].

## 2.8 Rotavirus screening

Only 91 of the stool samples collected had enough quantity for bacteria and rotavirus analysis. The 13 samples with low quantity of samples were excluded from the rotavirus screening. Rotavirus antigen detection was carried out by ELISA, using Human Rotavirus Antigen (RV-Ag)

ELISA KIT (EKHU-1933, Melsin Medical Co., Ltd-Changchun, China), following the manufacturer's guide (www.melsin.com/index.php?p=products_show&id=4308&lanmu=).

### 2.9 Statistical analysis

Data was analyzed using IBM Statistical Package for Social Science (SPSS) version 21.0. Results were summarized using frequency tables, percentages, mean and standard deviation. Pathogen positive and negative samples were compared in terms of gender, age-group, seasonal distribution, and clinical/risk factors using Chi-square with two-tailed significance. A p-value of 0.05 was considered statistically significant.

## 3. Results

### 3.1 Socio-demographic characteristics

Table 2 shows the socio-demographic characteristics of children having diarrhoea and that of the parent/guardian. Among 104 children recruited for this study, 67 (64.4%) were in the age group 0–11 months, 23 in age group 12–23, 9 and 4 in age group 24–35 and 48–60 respectively while only one child (0.96%) belong to age group 36–47 months. From the data collected, 61 (55.7%) were males and 43 (41.3%) of the enrolled children were females. Majority (62; 59.2%) of the parents/guardian have a family size of less than four, 41 (39.4%) use tap water as their source of drinking water and 42 (40.4%) of the parents/guardian had an average monthly income in the range of 10,000 ($22) and 30,000 naira ($65).

### 3.2 Prevalence of Rotavirus and bacterial single or multiple infection

At least one pathogen was detected from 63 (60.6%) out of the 104 children having gastroenteritis enrolled in the study. In multiple pathogen infection, the prevalence of RVA, DEC, *Shigella spp*., and *Salmonella spp*. was 45% (41/91), 23.1% (24/104), 21.2% (22/104) and 10.6% (11/104) respectively (Fig 1) while single infection prevalence was 22% (20/91), 5.8% (6/104), 5.8% (6/104), and 2.9% (3/104) respectively. There were 28 (44.4%) mixed infections among the diarrhoeic children. Specifically, double infection was detected in 22 (78.6%) children, triple infection in 4 (14.3%) and 2 (7.1%) of the children were infected with the four pathogens detected in the study (Table 3).

### 3.3 Highest pathogen prevalence observed in children less than 1 year

Of note is the fact that all the microbial aetiology of diarrhoea screened for in this study had their highest prevalence in children less than 1 year (Table 3), however there were no statistical relationship between age and pathogen detection (P = 0.233, 0.426, 0.880 and 0.634 for RVA, *E. coli*, *Shigella*, and *Salmonella* respectively).

Among the 61 males included in the study, 23 (37.7%), 15 (24.6%), 14 (23.0%), and 6 (9.8%) were positive for RVA, *E.coli*, Shigella, and Salmonella respectively, while 18 (41.9%), 9 (21.0%), 8 (18.6%), and 5 (11.6%) were detected among the 43 females. Notably, acute gastroenteritis (AGE) infection in our study was not sex-related (P> 0.05).

### 3.4 Diarrhoeagenic *Escherichia coli detected were mostly* Shiga Toxin *E. coli* (STEC)

Twenty-four DEC belonging to different *E. coli* pathotypes were detected in the study. Of the 24 DEC detected, majority of the isolates belong to the Shiga Toxin *E. coli* (STEC) while some were found to have cross bands of pathogenic genes. Nineteen (79.2%) of the isolates were

**Table 2. Socio-demographic characteristics of respondents.**

| Socio-demographic | Frequency (n = 104) | Percentage (%) |
|---|---|---|
| **Age of Child** | | |
| 0–11 months | 67 | 64.4 |
| 12–23 | 23 | 22.1 |
| 24–35 | 9 | 8.7 |
| 36–47 | 1 | 1.0 |
| 48–60 | 4 | 3.8 |
| **Sex of Child** | | |
| Male | 61 | 55.7 |
| Female | 43 | 41.3 |
| **Admission status** | | |
| Admitted | 26 | 25.0 |
| Outpatient | 79 | 75.0 |
| **Family Size** | | |
| Less than 4 | 62 | 59.6 |
| 4–6 | 37 | 35.6 |
| 7–10 | 4 | 3.8 |
| 11–15 | 1 | 1.0 |
| **Source of Drinking Water** | | |
| Stream | 2 | 1.9 |
| Well | 20 | 19.2 |
| Tap | 41 | 39.4 |
| Borehole | 16 | 15.4 |
| Sachet water | 18 | 17.3 |
| Bottled water | 7 | 6.7 |
| **Facility to Defecate** | | |
| Open ground | 3 | 2.9 |
| Pit toilet | 18 | 17.3 |
| Water system | 83 | 79.8 |
| **Washing of hand before feeding** | | |
| Yes | 89 | 85.6 |
| No | 4 | 3.8 |
| Sometimes | 11 | 10.6 |
| **Period of Breast Feeding** | | |
| 3 months exclusive breast feeding | 32 | 30.8 |
| 6 months exclusive breast feeding | 61 | 58.7 |
| Exclusive bottle feeding | 2 | 1.9 |
| Breast and bottle feeding | 9 | 8.7 |
| **Knowledge about RVA vaccine** | | |
| Yes | 19 | 18.3 |
| No | 85 | 81.7 |
| **Caregiver Level of Income in Naira** | | |
| Less than 10 thousand | 15 | 14.4 |
| 10–29 thousand | 42 | 40.4 |
| 30–49 thousand | 18 | 17.3 |
| 50–100 thousand | 9 | 8.7 |
| Above 100 thousand | 2 | 1.9 |
| No stable Income | 18 | 17.3 |

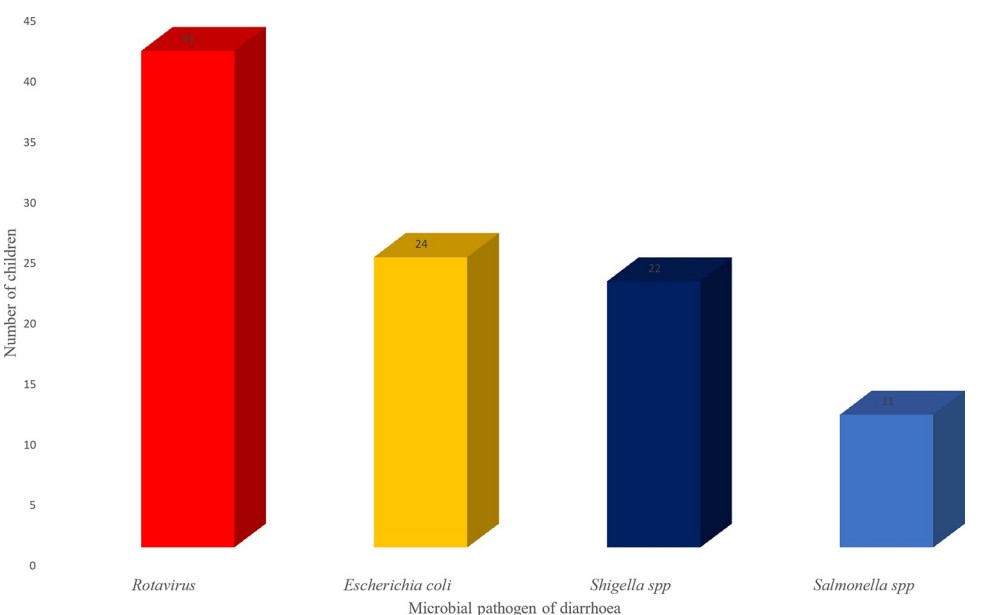

**Fig 1. Prevalence of Rotavirus and bacteria aetiology of diarrhoea in children.**

STEC, one isolate (4.2%) was found to be an EAEC while 4 (16,7%) isolates had a cross band of genes (stx1/stx2 and CVD342) belonging to STEC and EAEC as shown in Table 4.

## 3.5 Multidrug resistance observed in the DEC isolates

High multidrug resistance was observed in the DEC isolates recovered from our study. Diarrhoegenic *E. coli* isolates showed 100% resistance to the Cephalosporin (cefoxitin, cefuroxime sodium) and 95.8% resistance to the Sulfonamide (sulphamethoxazole/Trimethoprim). The highest percentage isolate susceptibility to antibiotics was associated with the Carbapenem class of antibiotics (Imipenem; 100%) followed by the Fluoroquinolones (Ciprofloxacin, 95.8%; Ofloxacin, 91.7%), Chloramphenicol (Chloramphenicol, 87.5%), and the least was

**Table 3. Distribution of single and multiple infections according to age in months.**

| Single (mono) infection | Age Group | | | | | |
|---|---|---|---|---|---|---|
| **Pathogen** | **0–11** | **12–23** | **24–35** | **36–47** | **48–60** | **Total** |
| RVA | 13 | 5 | 2 | 0 | 0 | 20 |
| DEC | 4 | 1 | 1 | 0 | 0 | 6 |
| *Shigella* | 5 | 1 | 0 | 0 | 0 | 6 |
| *Salmonella* | 1 | 1 | 1 | 0 | 1 | 3 |
| Total | 23 | 7 | 4 | 0 | 1 | 35 |
| Coinfection (multiple) | | | | | | |
| RVA + DEC | 5 | 1 | 1 | 0 | 0 | 7 |
| RVA + *Shigella* | 3 | 3 | 0 | 0 | 1 | 7 |
| RVA + *Salmonella* | 1 | 1 | 0 | 0 | 0 | 2 |
| DEC+ *Shigella* | 2 | 0 | 1 | 0 | 0 | 3 |
| DEC + *Salmonella* | 1 | 1 | 1 | 0 | 0 | 3 |
| *Shigella* + *Salmonella* | 1 | 0 | 0 | 0 | 0 | 1 |
| RVA + DEC + *Shigella* | 1 | 2 | 0 | 0 | 0 | 3 |
| RVA + DEC + *Shigella* + *Salmonella* | 2 | 0 | 0 | 0 | 0 | 2 |
| Total | 16 | 8 | 3 | 0 | 1 | 28 |

**Table 4. Distribution of diarrhoeagenic virulence genes in Faecal *Escherichia coli* from children.**

| Pathotypes | Number of subjects | Percentage (%) |
|---|---|---|
| STEC | 19 | 79.2 |
| EAEC | 1 | 4.2 |
| EAEC + STEC | 4 | 16.7 |
| ETEC | 0 | 0.0 |
| EPEC | 0 | 0.0 |
| EIEC | 0 | 0.0 |
| Total | 24 | 100 |
| **Virulence Genes** | | |
| *stx1 only* | 16 | 66.7 |
| *stx2 only* | 0 | 0.0 |
| *both stx1 and stx2* | 3 | 12.5 |
| *CVD342 only* | 1 | 4.2 |
| *CVD342 + stx1* | 3 | 12.5 |
| *CVD342 + stx2* | 0 | 0.0 |
| *CVD342 + stx1 + stx2* | 1 | 4.2 |

found in Tetracycline (Doxycycline, 33.3%). Fig 2 represents the susceptibility and resistance pattern of the isolates.

## 3.6 Clinical and environmental factors associated with rotaviral diarrhoea

Clinical and environmental factors associated with rotavirus were considered in the children screened for rotavirus (Table 5). Knowledge about RVA as the cause of paediatric diarrhoea

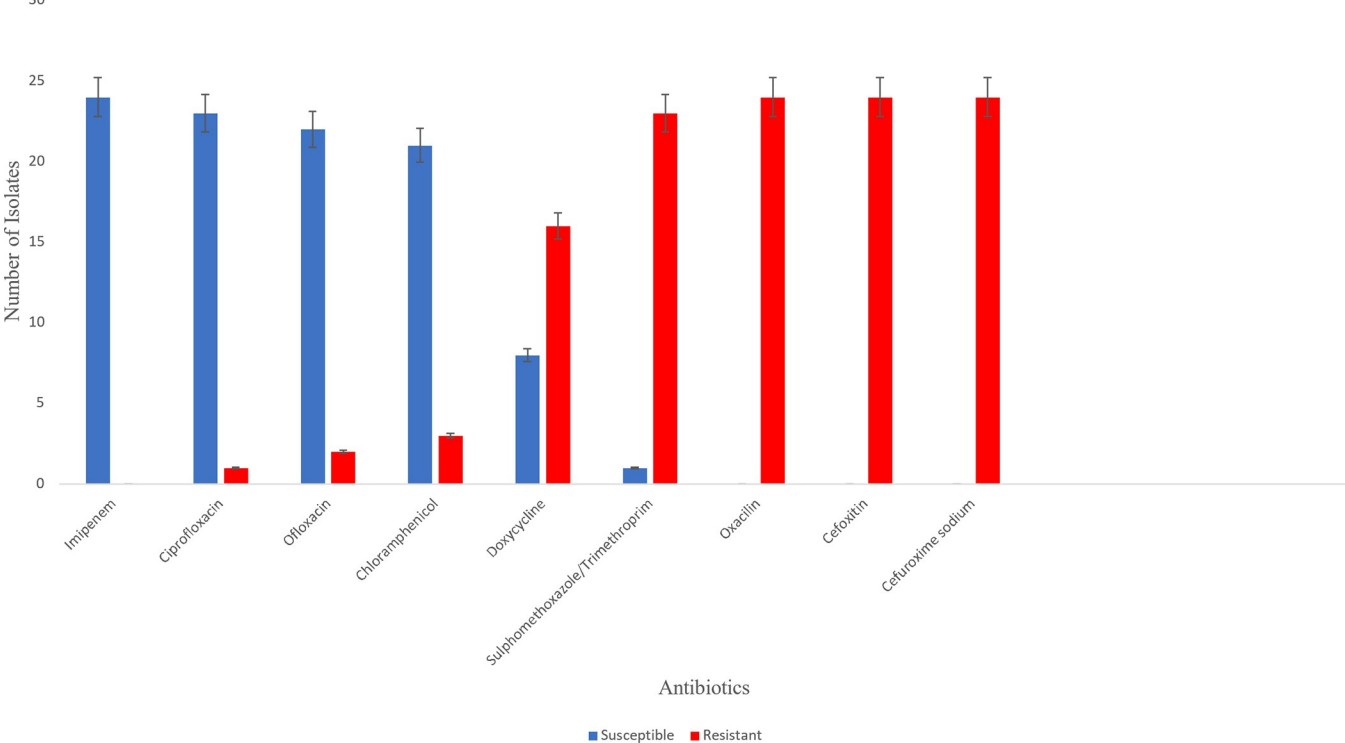

**Fig 2. Antibiotics sensitivity pattern of diarrhoeagenic *E. coli* (DEC).**

**Table 5. Clinical and environmental factors associated with Rotavirus infection.**

| Variables | Rotavirus infection | | $\chi^2$ | df | P value |
|---|---|---|---|---|---|
| | Rotavirus +ve (41) (%) | Rotavirus–ve (50) (%) | | | |
| **Anthropometric measurement** | | | | | |
| Less than 115mm | 8 (7.1) | 6 (42.9) | | | |
| Between 115 and 124mm | 7 (46.7) | 8 (53.3) | 13.556 | 6 | **0.035** |
| Between 125 and 134mm | 6 (27.3) | 16 (72.7) | | | |
| Greater than or equal to 135mm Total | 20 (50.0) | 20 (50.0) | | | |
| **Level of Income** | | | | | |
| High | 9 (32.1) | 19 (62.9) | 5.585 | 5 | **0.034** |
| Low Total | 32 (50.8) | 31 (49.2) | | | |
| **Knowledge about RVA** | | | | | |
| Yes | 35 (50.7) | 34 (49.3) | 3.706 | 1 | 0.054 |
| No Total | 6 (27.3) | 16 (72.7) | | | |
| **Source of Drinking Water** | | | | | |
| Stream | 1 (50.0) | 1 (50.0) | | | |
| Well | 8 (47.1) | 9 (52.9) | | | |
| Tap | 16 (43.2) | 21 (56.8) | 1.141 | 5 | 0.950 |
| Borehole | 7 (50.0) | 7 (50.0) | | | |
| Sachet water | 5 (35.7) | 9 (64.3) | | | |
| Bottled water | 4 (57.1) | 3 (42.9) | | | |
| Hand washing Practice Always Sometimes Rarely | 36 (47.4) 3 (27.3) 2 (50.0) | 40 (52.6) 8 (72.7) 2 (50.0) | 0.907 | 2 | 0.636 |
| **Toilet facility** | | | | | |
| Open ground | 1 (50.0) | 1 (50.0) | | | |
| Pit toilet | 6 (40.0) | 9 (60.0) | 0.198 | 2 | 0.906 |
| Water system | 34 (45.9) | 40(54.1) | | | |
| **Admission Status** | | | | | |
| Admitted | 9 (36.0) | 16 (64.0) | 1.142 | 1 | 0.285 |
| Outpatient | 32 (48.5) | 34 (51.5) | | | |
| **6 months exclusive breastfeeding** | | | | | |
| Yes | 30 (49.2) | 31 (50.8) | 3.306 | 3 | 0.347 |
| No | 11 (36.7) | 19 (63.3) | | | |
| Stool Description | | | | | |
| Watery | 22 (52.4) | 20 (47.6) | | | |
| Loose | 9 (32.1) | 19 (67.9) | 2.724 | 1 | 0.099 |
| Mucus | 10 (58.8) | 7 (41.2) | | | |
| Bloody | 0 (0.0) | 4 (100) | | | |
| **Duration of Stooling** | | | | | |
| Less than 1 day | 10 (62.5) | 6 (37.5) | | | |

(*Continued*)

**Table 5.** (Continued)

| Variables | Rotavirus infection | | $\chi^2$ | df | P value |
|---|---|---|---|---|---|
| 1–3 days | 20 (37.0) | 34 (63.0) | 0.668 | 3 | 0.881 |
| 4 days and above | 11 (52.4) | 10 (47.6) | | | |
| **Associated Fever** | | | | | |
| Yes | 28 (50.0) | 28 (50.0) | 1.438 | 1 | 0.230 |
| No | 13 (37.1) | 22 (62.9) | | | |
| **Associated Dehydration** | | | | | |
| Yes | 21 (51.2) | 20 (48.8) | 1.145 | 1 | 0.284 |
| No | 20 (40.0) | 30 (60.0) | | | |
| **Skin Characteristics** | | | | | |
| Sunken eyes | 6 (50.0) | 6 (50.0) | | | |
| Loss of skin turgor | 2 (40.0) | 3 (60.0) | 0.668 | 3 | 0.881 |
| Both | 4 (57.1) | 3 (42.9) | | | |
| None | 29 (43.3) | 38 (56.7) | | | |

was high among the parent/guardian. Specifically, 75.8% (69/91) of the parents have knowledge about the virus. The data shows that RVA positive stools were not bloody (0%; n = 0/4) but could be watery (52.4%; n = 22/42), loose (32.1%; n = 9/28), or mucoid (58.8%; n = 10/17).

Rotavirus A positive samples were common in children whose parents have low level of income (78.0%; n = 32/41). In our study, RVA infection was statistically associated with level of income of the parent and child Anthropometric measurement (mid-upper arm circumference measurement), indicative of low nutritional status (malnutrition) in the children (P = 0.034 and 0.035 respectively). There was no statistical relationship (P> 0.05) between RVA incidence and the environmental factors considered in this study (Table 5).

### 3.7 Rotavirus infection occurred only in the dry season

In this study, the two distinct seasonal peaks in Nigeria were characterized with different microbial aetiology of diarrhoea. A consistent rise in rotavirus infection was observed towards the dry season, with sharp increase in January and February (Fig 3). About 75% of RVA infection recorded in this study was from samples collected in January and February. The causative organism from bacteria origin was more prominent during the rainy season which falls within April and September.

### 4. Discussion

This study reports a high prevalence of microbial gastroenteritis among children less than 5 years in Ile-Ife, Osun State, Nigeria. To the best of our knowledge, there is no recent single study report on bacterial and viral aetiology of diarrhoea among children in Nigeria. The only available study is the study by Ogunsanya et al., [20] about two decades ago with a prevalence of 74.9% microbial aetiology of diarrhoea in children. This shows that diarrhoea is still a major problem in the country with a prevalence of 60.6% in our study.

Although, there are no recent studies carried out in Nigeria with reports of the prevalence of viral and bacterial diarrhoea in a single study population with which to compare the result of this study, the findings of this study is consistent with recent reports from other developing countries like Niger, Burkina Faso, Nicaragua, India, and Turkey where viral and bacterial

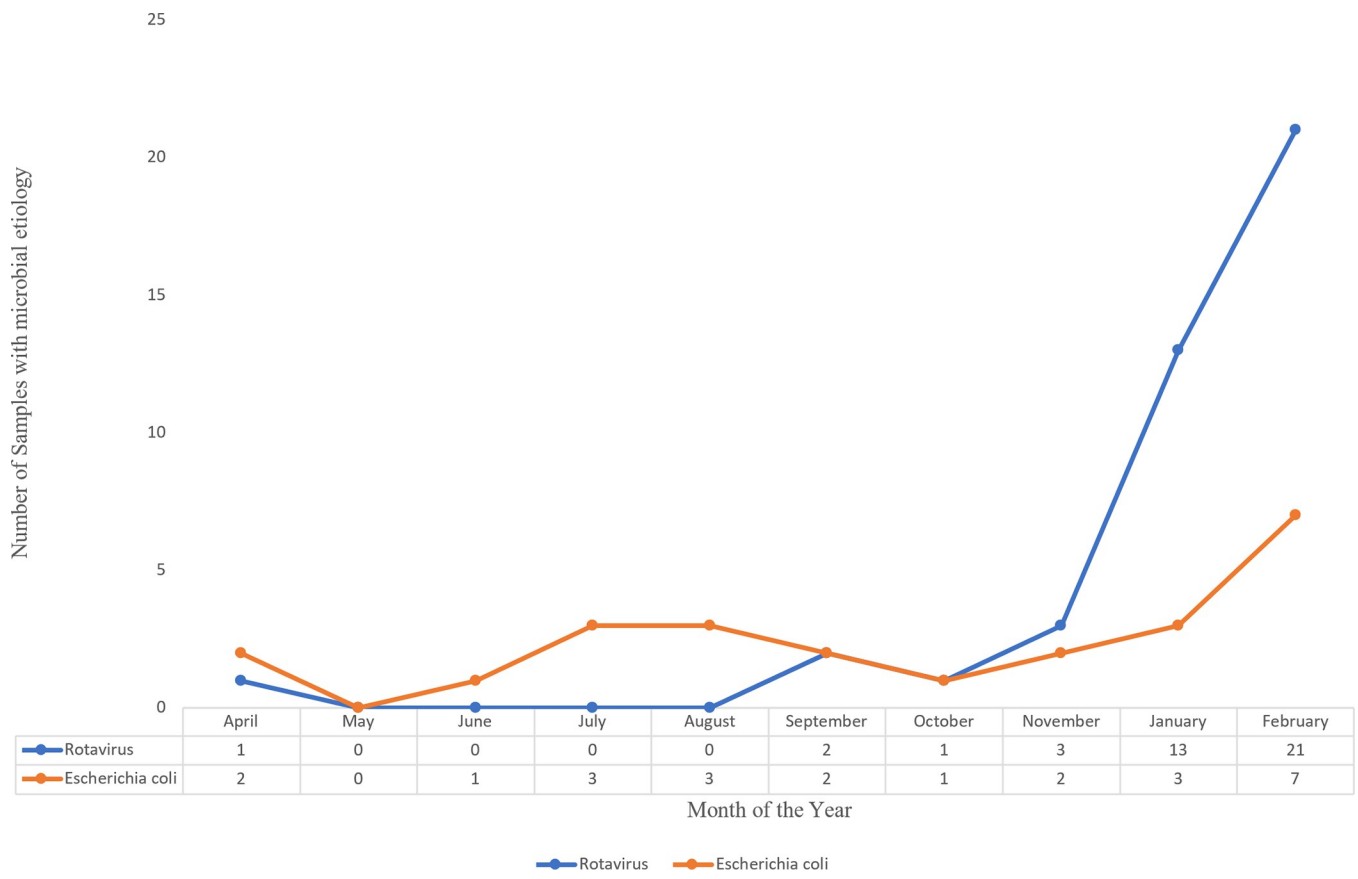

**Fig 3. Seasonal variation in viral and bacterial aetiology of diarrhoea.**

gastroenteritis prevalence of 70%, 64%, 61.1%, 60%, and 59.2% were reported respectively [21–24]. The high prevalence of diarrhoea from the microbial origin in children associated with this region might be a result of low-middle income-earning and a high number of children per household, making the provision of safe drinking water, good sanitation facilities, and balanced diet difficult.

Children living in poor households have been reported to be more vulnerable to diarrhoea than their wealthy counterparts [25]. Our study is in line with this study, showing that children from high income parents are less likely to have RVA diarrhoea compared to their counterparts from low income parents who make up 78% of the RVA positive children with malnutrition.

Rotavirus infection in this study was observed to have the highest prevalence among the different microbial aetiology of childhood diarrhoea screened for. This result supports previous findings [4,21,26,27] with high rotaviral infection compared to other microbial aetiology of diarrhoea among children <5 years. This might not be unconnected with the fact that majority of children are not vaccinated against rotavirus infection and the spread of the virus is aided by environmental factors in their community. Till date, rotavirus vaccine is yet to be introduced into the free Nigeria national immunization program but is still being paid for by parents who can afford it. Low prevalence of RVA has been reported by countries where RVA vaccine is fully introduced into their national immunization scheme [22,28–32]. As suggested by Japhet et al., [33], Nigeria needs to consider including rotavirus vaccine into the free immunization given to children.

Of the different DEC identified in this study, Shiga toxin-producing *E. coli* (STEC) was the most common sub-pathotype of DEC detected, corroborating the report of Odetoyin et al., [18] that STEC is the prevalent pathotype of DEC in children but deviates from Onanuga *et al.*, [34] and Scholar et al., [35] who reported EAEC as the most common DEC pathotype in children. The difference in result could be as a result of the different geographical study locations. Interestingly, our study and that of Odetoyin [18] were carried out in the South-Western part of Nigeria while the studies of Onanuga [34] and Scholar [35] with similar results but expectedly different from ours, were carried out in the Northcentral part of the country. The prevalence of STEC in the environment poses a risk of having hemolytic uremic syndrome and hemorrhagic colitis which are known to be severe STEC community disease [18].

Rotavirus and bacteria coinfection was reported among diarrhoeic children in this study, with rotavirus and DEC coinfection being the most observed, rotavirus with *Shigella* coinfection was the next bacteria coinfection between *Shigella and Salmonella* was observed as the least co-infection in the study. Since there is no recent study in the country that has investigated the aetiology of diarrhoea from viral and bacterial origin to compare the result with, this result is similar to previous findings in developing countries where the combined infection between viral and bacterial etiologies of diarrhoea has been reported [21,22,24,36]. However, co-infection had no statistical significant on the severity of diarrhoea (P = 0.523).

Of the diarrhoeic children with at least one pathogen, a higher prevalence of diarrhoea infection was associated with children less than 1 year. This finding corroborates previous studies carried out in Nigeria and neighbouring developing countries [27,33,37], in which majority of the diarrhoea infection was reported in children less than one year. This might most likely be due to the low level of immunity in the 0 to 11 months age group [5].

In this study, viral and bacterial gastroenteritis were more prevalent among males than females for all the pathogen tested, although viruses and bacteria detection were not sex related. Previous studies in Nigeria were either on bacteria or viruses singly, hence, there is no baseline data for comparison of these agents with sex. However, studies on viral AGE infection in Nigeria and elsewhere have reported that no statistical significance exist between detection of enteric viruses and the sex of a child [27,38,39].

Diarrhoea is classified as mild, moderate, and severe based on the presence or absence of dehydration, sunken eyes, and loss of skin turgor [40]. In this study, different environmental and risk factors were observed to contribute to diarrhoea prevalence in the children. Malnutrition has been said to be directly related to diarrhoea [41]. In agreement with this report, findings from this study shows that diarrhoea was more severe in children that are malnourished (i.e children with low anthropometric measurement) compared to their other counterpart (P = 0.035). Reduced burden of diarrhoea in children under the age of 5 years can result from addressing malnutrition or improving children's nutritional status [42].

With regards to seasonal distribution of pathogen, gastroenteritis caused by rotavirus began from September, with a significant seasonal peak between January and February. In South-western Nigeria where this study was carried out, these months correspond to the dry, cold season of the year [43]. As reported by Abebe [44], Nigeria has two main seasons in a year, namely wet (rainy) and dry (dry cool months between September to February) which tallies with that of many other tropical countries [45]. The finding in this study was similar to a recent report from Nigeria [27] where the peak for rotavirus gastroenteritis was observed between December and March.

Since most diarrhoea in children is associated with rotavirus infection especially in the dry cool months, childhood diarrhoea can or should be easily managed without the intervention of antibiotics. Therapy using the antimicrobial drug should only be recommended when

diarrhoea is invasive or persistent which most likely might be as a result of coinfection between viral and bacterial aetiology [18,46,47].

In this study, multidrug resistance was observed in the DEC isolates recovered from our study, corroborating reports of high percentage resistance of DEC isolates to various classes of antibiotics from Nigeria and neighbouring countries [18,37,48]. All strains of DEC isolates in our study showed 100% resistance to the cephalosporin (cefoxitin, cefuroxime sodium) class of antibiotics while all (100%) were susceptible to the Carbapenem class of antibiotics (Imipenem). The high sensitivity of isolates to imipenem observed in this study agrees with the 100% susceptibility reported by Oladipo [49]. This could be linked with less abuse of this drug, as it is mostly administered intravenously, restricting the prescription to medical practitioners, thereby limiting self-medication. We therefore suggest imipenem and ciprofloxacin as good option of antibiotics to combact persistent and invasive diarrhoea within our locality.

The limitations of this study was the inability to collect samples in some parts of December 2019 and March 2020 due to the outbreak of the Lassa fever virus in Nigeria and the outbreak of the global coronavirus pandemic which brought about restrictions in gaining access to the hospital and made parents to avoid coming to the hospital. Also, we were unable to carryout RVA molecular analysis of the stool samples, however, previous studies have reported correlation between ELISA and real time PCR results [50–54] and low Ct-values (high viral load) is being related to more severe disease, hence the study is not in any way impaired.

## 5. Conclusion

This study reports the burden of rotavirus and bacterial as aetiology diarrhoea and the associated risk factors in Nigeria after two decades of lack of report, showing a high microbial acute gastroenteritis among children 0–5 years with RVA as the most prevalent pathogen in children less than one year. Other findings include rotavirus seasonal peaks, and high multidrug resistance. There is a need to concentrate on viral gastroenteritis research, reduce indiscriminate use of antibiotics and include free rotavirus vaccine into the Nigerian National immunization scheme to reduce diarrhoea burden in Nigeria and elsewhere.

## Supporting information

**S1 File.**
(DOCX)

## Acknowledgments

We are very grateful with sincere appreciation to all parents and children who participated in the study.

## Author Contributions

**Conceptualization:** Temiloluwa Ifeoluwa Omotade, Margaret Oluwatoyin Japhet.

**Data curation:** Temiloluwa Ifeoluwa Omotade, Toluwani Ebun Babalola, Chineme Henry Anyabolu, Margaret Oluwatoyin Japhet.

**Formal analysis:** Temiloluwa Ifeoluwa Omotade, Toluwani Ebun Babalola, Chineme Henry Anyabolu, Margaret Oluwatoyin Japhet.

**Investigation:** Temiloluwa Ifeoluwa Omotade, Toluwani Ebun Babalola, Chineme Henry Anyabolu, Margaret Oluwatoyin Japhet.

**Methodology:** Temiloluwa Ifeoluwa Omotade, Margaret Oluwatoyin Japhet.

**Project administration:** Temiloluwa Ifeoluwa Omotade, Toluwani Ebun Babalola, Chineme Henry Anyabolu, Margaret Oluwatoyin Japhet.

**Resources:** Temiloluwa Ifeoluwa Omotade, Toluwani Ebun Babalola, Chineme Henry Anyabolu, Margaret Oluwatoyin Japhet.

**Software:** Temiloluwa Ifeoluwa Omotade, Margaret Oluwatoyin Japhet.

**Supervision:** Margaret Oluwatoyin Japhet.

**Validation:** Temiloluwa Ifeoluwa Omotade, Toluwani Ebun Babalola, Chineme Henry Anyabolu, Margaret Oluwatoyin Japhet.

**Visualization:** Temiloluwa Ifeoluwa Omotade, Margaret Oluwatoyin Japhet.

**Writing – original draft:** Temiloluwa Ifeoluwa Omotade, Margaret Oluwatoyin Japhet.

**Writing – review & editing:** Temiloluwa Ifeoluwa Omotade, Toluwani Ebun Babalola, Chineme Henry Anyabolu, Margaret Oluwatoyin Japhet.

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
