## [Decision Letter · Decision Letter 0]

5 Oct 2022

PONE-D-22-22211Rotavirus and bacterial diarrhoea among children in Ile-Ife, Nigeria: burden, clinical/environmental factors and seasonality.PLOS ONE

Dear Dr. Japhet,

Thank you for submitting your manuscript to PLOS ONE. After careful consideration, we feel that it has merit but does not fully meet PLOS ONE’s publication criteria as it currently stands. Therefore, we invite you to submit a revised version of the manuscript that addresses the points raised during the review process.

We look forward to receiving your revised manuscript.

Kind regards,

Tacilta Nhampossa

Academic Editor

PLOS ONE

Journal Requirements:

 Whilst you may use any professional scientific editing service of your choice, PLOS has partnered with both American Journal Experts (AJE) and Editage to provide discounted services to PLOS authors. Both organizations have experience helping authors meet PLOS guidelines and can provide language editing, translation, manuscript formatting, and figure formatting to ensure your manuscript meets our submission guidelines. To take advantage of our partnership with AJE, visit the AJE website (http://aje.com/go/plos) for a 15% discount off AJE services. To take advantage of our partnership with Editage, visit the Editage website (www.editage.com) and enter referral code PLOSEDIT for a 15% discount off Editage services. If the PLOS editorial team finds any language issues in text that either AJE or Editage has edited, the service provider will re-edit the text for free.

4. We note you have included a table to which you do not refer in the text of your manuscript. Please ensure that you refer to Table 2 in your text; if accepted, production will need this reference to link the reader to the Table.

Additional Editor Comments:

This is a very interesting study. However, there are several issues in the manuscript that require further elaboration or at least clarification. To ensure the Editor and Reviewers will be able to recommend that your revised manuscript is accepted, please pay careful attention to each of their comments. This way we can avoid future rounds of clarifications and revisions, moving swiftly to a decision.

Reviewers' comments:

Reviewer's Responses to Questions

**Comments to the Author**

1. Is the manuscript technically sound, and do the data support the conclusions?

Reviewer #1: Yes

Reviewer #2: Partly

2. Has the statistical analysis been performed appropriately and rigorously? 

Reviewer #1: Yes

Reviewer #2: No

3. Have the authors made all data underlying the findings in their manuscript fully available?

Reviewer #1: Yes

Reviewer #2: No

4. Is the manuscript presented in an intelligible fashion and written in standard English?

Reviewer #1: Yes

Reviewer #2: No

5. Review Comments to the Author

Reviewer #1: The manuscript is well written and easy to read. Since Nigeria recently introduced rotavirus vaccination in the national immunisation program, the information in this manuscript contributes to information regarding rotavirus frequencies befor the introduction (even through Rotarix has been available for some time). Although seasonal variations of rotavirus and to some extent of bacterial infections are already known, this information may be useful to clinicians in Nigeria. I assume that there are limited resources for diagnostics and therefore the seasonal variation can be informative as analyses can be prioritised depending on season.

Material and methods:

1.Minor Comment: Diarrhoea is spelled wrong on line two in chapter 2.1.

2. Unfortunately the instructions for the Rota virus Ag ELISA couldn't be found on the webb of Melsin Medical. Do you know the specificity and sensitivity of the test, compared to other diagnostic tests, like PCR, for rotavirus? I completely understad that you have used what is available but I suspect that you would have found even more cases with a more sensitive method.

Result:

1.Table 1, at first sign, it is difficult to adress that the numbers (in line of pathogens) belongs to age groups. it will be easier to read if the age-groups are more marked, maybe on line up.

2. You have showed that there are different pathogens circulating during different seasons, which makes it difficult to relate to the information in Table 2. Many of the presented cases in the rotavirus negative group have another pathogen, in another season, that likely affects their health. To really show whether there are differences in, for example, dehydration, healthy controls should be included, alternatively the comparison should be made with negative cases within the same season. It would have been interesting to see the same table divided in seasons. Dry and rain maybe.

Discussion: In this manuscript the co-infections are pointed out int result and tables. I miss a comment in the discussion about their effect. Will a co-infection give more symptoms or is it one pathogen that causes disease?

Reviewer #2: Dear Academic Editor, PLOS ONE

Thank you for your invitation to review the manuscript “Rotavirus and bacterial diarrhoea among children in Ile-Ife, Nigeria: burden, clinical/environmental factors and seasonality”. The manuscript describes a basic hospital investigational of diarrhoea. There are several issues in the manuscript that require further elaboration or at least clarification. Please the comments attached.

6. PLOS authors have the option to publish the peer review history of their article (what does this mean?). If published, this will include your full peer review and any attached files.

Reviewer #1: No

Reviewer #2: No

---

## [Author Response · Author response to Decision Letter 0]

9 Dec 2022

REVIEWER 1

1.Minor Comment: Diarrhoea is spelled wrong on line two in chapter 2.1

Response: We appreciate the reviewer for the observation of this spelling error. Diarrhoea has been spelt correctly.

2. Unfortunately the instructions for the Rota virus Ag ELISA couldn't be found on the webb of Melsin Medical. Do you know the specificity and sensitivity of the test, compared to other diagnostic tests, like PCR, for rotavirus? I completely understad that you have used what is available but I suspect that you would have found even more cases with a more sensitive method.

Response: We greatly appreciate the reviewer for this observation, Melsin have been contacted and they have updated their website to include the instruction for the Rotavirus antigen ELISA kit. Although, the specificity and sensivity of the kit is not included on their website, our result is broadly in line with other rotavirus ELISA studies from the region using different ELISA kit which recorded similar prevalent value (30.7% to 47.6%) suggesting that there was no significant bias. Japhet et al., 2018; Tagbo et al., 2019

Result:

3. Table 1, at first sign, it is difficult to adress that the numbers (in line of pathogens) belongs to age groups. it will be easier to read if the age-groups are more marked, maybe on line up.

Response: Thank you very much for this observation. The age groups have been marked for easy read. 

4. You have showed that there are different pathogens circulating during different seasons, which makes it difficult to relate to the information in Table 2. Many of the presented cases in the rotavirus negative group have another pathogen, in another season, that likely affects their health. To really show whether there are differences in, for example, dehydration, healthy controls should be included, alternatively the comparison should be made with negative cases within the same season. It would have been interesting to see the same table divided in seasons. Dry and rain maybe.

Response: The table refered to above is now table 5 due to the addition of some other tables. Most of the rotavirus negative group included in the analysis for table 5 do not have another pathogen in another season. Figure 3 showed the trend of the pathogen accross the sampling period.

5. Discussion: In this manuscript the co-infections are pointed out int result and tables. I miss a comment in the discussion about their effect. Will a co-infection give more symptoms or is it one pathogen that causes disease?

Response: There was no significant statistical relationship between the severity of diarrhoea and coinfection. This has been stated in the discussion (P= 0.523).

REVIEWER 2

1. Title

Consider to review the manuscript title, because it does not reflect what has been done as methodology and main findings, to be more concise and suggestive.

Response: Thank you for the suggestion, however, we think it necessary to retain the title but we adjusted the methodology and main findings to align with the title.

2. Abstract

The abstract is too long, needs to be shorten and synthetized. Additionally, seems to be missing the justification for conducting the study. There are many statements, which needs to be reviewed:

 “Pathogens were detected using ELISA, conventional cultural technique and PCR. Escherichia coli (E. coli) isolates were pathotyped”: Was the pathotyping method different from PCR? Please, specify which pathogen was detected by which technique.

“etiology”: Please standardize, sometimes it appears aetiology.

“E. coli. was the most common bacteria etiology with high multidrug resistance in the DEC isolates”: Needs to be corrected. Considering that E. coli can also be commensal, the authors should only consider the frequency of DEC and classification by each pathotype rather the overall frequency of E. coli isolation.

“Variation in seasonal peaks of viral and bacterial etiology of diarrheoa”: This should be interpreted with caution because the figure of seasonality (Fig. 3) only describes rotavirus and E.coli. 

“The E. coli pathotypes detected were Shiga Toxin E. coli (STEC), enterohaemorrhagic E. coli (EHEC) and enteroaggregative E. coli (EAEC)”. Why the author considered STEC and EHEC as distinct pathotypes. What is the difference between both? Your consideration for the papers below regarding the classification for this pathotype.

Delannoy, S., Beutin, L., and Fach, P. (2013). Discrimination of Enterohemorrhagic Escherichia coli (EHEC) from Non-EHEC Strains Based on Detection of Various Combinations of Type III Effector Genes. J. Clin. Microbiol. 51, 3257–3262. doi: 10.1128/JCM.01471-13.

Orth, D., and Wurzner, R. (2006). What Makes an Enterohemorrhagic Escherichia coli? Clin. Infect. Dis. 43, 1168–1169. doi: 10.1086/508207.

Response: The length of the abstract has been reduced to 300 words without watering down the content. The justification for the study is included in the 300 words. 

The statement “Pathogens were detected using ELISA, conventional cultural technique and PCR. Escherichia coli (E. coli) isolates were pathotyped” has been re-written, with pathogen specific technique(s) specified. “Aetiology” has been replaced with “etiology throughout the manuscript.

“E. coli. was the most common bacteria etiology with high multidrug resistance in the DEC isolates”. We were reporting for only DEC isolates, but we have reconstructed the statement for clarity.

“The E. coli pathotypes detected were Shiga Toxin E. coli (STEC), enterohaemorrhagic E. coli (EHEC) and enteroaggregative E. coli (EAEC)”. Why the author considered STEC and EHEC as distinct pathotypes……….. pathotype”.

Response: We thank the reviewer for the contribution and the suggested publications for consideration. Since STEC is a subgroup of EHEC, we agree with the reviewer, hence STEC has been expunged.

3. Introduction

The global burden of diarrhoea should be updated with recent data, the authors present data from 5 years ago ((WHO 2017; UNICEF 2018). 

Response: We appreciate the reviewer for the observation, more recent data from WHO has been included.

Statements: 

“Considering the immaturity of children’s immune system, these poor living conditions increase the vulnerability of children younger than 5 years to diarrhoea when exposed to pathogens that cause diarrhoea”: Please review the sentence. “(rotaviruses, caliciviruses, noroviruses, and sapoviruses)”: Noroviruses and sapoviruses are Caliciviruses.

Response: This was an oversight and has been corrected appropriately. 

“or having contact pathogen carriers”: Please clarify what does this statement means.

Response: Thank you for the observation. The statement has been removed because of its redundancy and lack of further meaning to the sentence.

“Western states due to the level of enlightenment”: What does enlightenment means and relation to pathogens?

Response: We are appreciative of your logical question; the level of enlightenment has been replaced with a level of heightened hygiene as the level of enlightenment did not properly convey what we intended.

“Nigeria being a tropical country has two main seasons: the dry season which lasts from October until around April (or March in some parts of the country), determined by high temperatures and low humidity and the wet or rainy season which is for the remaining months of the year”: What is the relevance to bring Nigeria season in the introduction? It should be moved to methodology.

Response: We appreciate the reviewer for this comment, Nigeria season has been removed from introduction to discussion

The authors should consider mentioning in the justification that there is lack of data regarding diarrhoea aetiology in Nigeria and that the study is bringing some data to fill this gap

“Escherichia coli (ETEC), enteropathogenic Escherichia coli (EPEC), Shigella species, Samonella species, Campylobacter jejuni), and protozoa or parasites (Entamoeba histolytica, Giardia lambia)”. Correct the taxonomy for Salmonella spp.

Response: We have included the justification of this study and the taxonomy of Salmonella spp. has been corrected as pointed out by the reviewer.

4. Materials and Methods

In general authors should review profoundly the study design, there is some key information missing, like description of inclusion criteria, definition of acute gastroenteritis, sample collection and storage.

Specifically, in: 

2.1. Study design: “Informed consent was taken from the caregivers of all the study participants, and children who have taken antibiotics within 10 days of presentation were excluded from the study. Why did the authors considered excluding children who took antibiotics in just 10 days and not more? How did the authors ensure that the caregiver recognized whether the child received an antibiotic or other unrelated medication?

Response: The statement has been rewritten, and the definition of acute gastroenteritis has been included. The inclusion criteria have also been clearly stated.

2.2 Ethics”

Ethics Approval: “Written informed consent was taken from caregivers. A semi-structured interviewer administered questionnaire was administered to caregivers of study participants, detailing relevant sociodemographic information, clinical and environmental factors”: This should not be part of the ethical section.

Response: The statement has been removed.

Why did the authors not perform AST for Salmonella and Shigella?

Response: This is due to lack of funding, research in Nigeria is mostly self-funded.

DNA Extraction: “The DNA of the bacteria”: Should not be E.coli colonies? As PCR was only performed for its pathotypes.

Response: The sentence has been changed to E. coli colonies. PCR was not only done for the pathotyping, confirmation of E. coli isolates was also carried out by PCR.

“The plates were incubated for 24 hrs. aerobically at 37 °C. Distinct colonies were picked

and identified by standard biochemical tests.” What was the procedure for those cultures with no growth within 24 hrs, the plates were reincubated?

Response: Culture with no growth was recorded as negative after 24 hours of incubation. It was an oversight not to have included the procedure for cultures with no growth. This information has been added to the manuscript.

“The isolated organisms suspected to be E. coli by their cultural and biochemical characteristics

were confirmed as E. coli by polymerase chain reaction (PCR) using primers specific to E. coli

16S rRNA gene, adopting the procedure described by Mamun et al. (2016).” This reference is not the appropriate. Please consider to add appropriate reference. In addition, specify a little bit more about the primers and protocols used.

Response: Table 1 showing details of the primers and protocol used for E.coli confirmation in this study has been included in the manuscript and appropriate reference cited .

“All the E. coli isolates were screened for virulence genes of five different pathotypes of diarrhoeagenic E. coli, including enterotoxigenic E. coli (ETEC) and enteroinvasive E. coli (EIEC), enteropathogenic E. coli (EPEC), enteroaggregative E. coli (EAEC), and enterohaemorrhagic E. coli (EHEC) as previously described by Aranda et al. (2004) with slight modifications.” Mention the targets used for DEC detection. Which modifications were made? Describe more.

Response: The targets for DEC detection and the virulence genes screened for have been included. We have removed with slight modifications since these where only in volumes and do not affect concentrations or method

“The disc diffusion method was used for the antibiotic susceptibility testing of DEC isolates on

Mueller Hinton agar. Commonly used and available antibiotics (Oxoid Ltd, UK) impregnated…….” Which antibiotics were used for AST. Are they common used in Nigeria? Why the authors only tested for 6 antibiotics?

Response: All the antibiotics tested (chloramphenicol (30µg), ciprofloxacin (5µg), ofloxacin (5µg), cefuroxime (30µg), imipenem (10µg), cefoxitin (30µg), doxycycline (30µg), and sulphamethoxazole/ Trimethroprem (25µg)) have been included. We tested 8 different commonly used antibiotics in Nigeria from six different classes of antibiotics (Cephalosporin, Sulphonamide, Carbapenem, Fluoroquinolones, Chloramphenicol, and Tetracycline).

“After incubation, the plate was well aspirated and washed 5 times using the wash….”The washing was manual or automatized?

Response: The washing was done manually.

“Finally, 50 μl of stop solution was added to all the wells and the result read on ELISA plate reader at 450”. Provide the equipment description.

 Rotavirus Screening: Please if ELISA steps are described in the manufacture'r protocol, please resume.

Response: Because of this comment, we have reviewed rotavirus screening procedure in the manuscript to just a brief description, directing readers to manufacturer’s protocol. We thank the reviewer for this important counsel.

Statistical Analysis: Please review the meaning of the statistical package (SPSS).

“Student T-test and Chi-square”: P value is used to compare and test the significance of your hypothesis and not to draw conclusions but to suggest what may be the factors.

Response: There was a typographical error in the meaning of SPSS which has been corrected. 

5. Results

In general, the authors should consider bringing a table describing the general characteristics of the population before presenting specific analysis, also adding information related to the virulence factors. Additionally, all the subtitles should be reviewed, authors describe them as if they were giving a conclusion or the main finding, ignoring the whole data in the section. All the figure quality should be improved with programs with better resolution. In all the table and images should be added the place, kind of population and study period. There are many results without specifying the correspondent table/figure, please review. There are also so many redundancies.

 “Rotavirus more prevalent than bacterial in mono or multiple infection”: Please review the numbers and percentages, sometimes it appears n (%) without a denominator while sometimes both. Review the terms: single co-infections, double, triple, four, which the pathogens are not specified in the section.

“Majority of the pathotyped isolates belong to the Shiga Toxin E. coli (STEC)”: Please specify or explain more what does cross bands means.

Multidrug resistance observed in the DEC isolates: Please indicate how many isolates were tested ans? Also present the resistance according to the pathotypes. 

Clinical and environmental factors associated with rotavirus diarrhoea: “samples with quantities not enough for both bacterial and viral screening were excluded”: This should have been mentioned in the methods section.

Table 2: The clinical and risk factors should be presented separately. Specify if the numbers in brackets are numbers or percentages.

Rotavirus infection occurred only in the dry season: “Samples were not collected in December 2019 and March 2020 due to the outbreak of the Lassa fever virus in Nigeria and the outbreak of the global coronavirus pandemic which brought about restrictions in gaining access to the hospital and made parents to avoid coming to the hospital”: This information should have been described in the method section and authors should adjust the study period. This information would also be cited as study limitation.

“At least one pathogen was detected from 63 (60.6%) of the children having gastroenteritis from

the one hundred and four children enrolled in this study.” In the abstract the authors mentioned that only 90 children had sufficient samples for rotavirus screening. Now the author considered all the 104 children. Please check.

The AST data should be complemented with detection of resistant determinants.

“In multiple pathogen infection, the prevalence of RVA, E.coli, Shigella spp., and Salmonella spp. was 45% (41/91), 23.1% (24/104), 21.2% (22/104) and 10.6% (11/104) respectively……….”

When the authors mentioned E. coli are referring to all DECs or all E. coli isolated by conventional microbiology? The authors should only consider DECs for analysis.

“….genes. Sixteen (69.6%) of the isolates were STEC, 2 (8%) were EHEC, one isolate (4%) was……” What is the difference between STEC and EHEC?

“Multidrug resistance observed in the DEC isolates”. All this section should be rewritten, there are so many redundancies.

“Multidrug resistance (the resistance of an organism to 3 or more different classes of antibiotics)”. This sentence should be moved to methodology.

 “Table 1: distribution of Single and Multiple infections according to age.”Consider to include %s.

Response: We appreciate the author for the constructive criticism and evaluation of this section. A Table which is now ‘Table 2’ that generally describes the characteristics of the population has been added. All the results have been reviewed and the corresponding figures/tables have been adequately specified. The numbers and percentages have been reviewed and the appearance is unified. Table 3 which is on the distribution of virulence genes has been included.

Only the DEC was considered for the analysis, the places that will have E. coli have been replaced with DEC accordingly. 

The definition of multidrug resistance as adopted in this study has been included in the methodology as pointed out by the reviewer.

6. Discussion 

Overall, the discussion is too long, the authors should summarize the text and focus on their results first, before referencing similar findings. 

Minor comments:

Please give more recent publication for comparisons of diarrhoea aetiology in Nigeria, (Ogunsanya et al., 1994). The authors should even use publication referring single diarrhoea pathogens detection.

Response: We appreciate the reviewer for the point raised to contribute positively to our study. However, from our limited literature search, there are no recent similar studies in the country with which we can compare our combined prevalence with. Publication on the single prevalence of diarhhoea pathogen was used in the study which includes Scholar 2020; Tagbo et al., 2019; Japhet et al., 2018; and Odetoyin et al., 2016.

“Niger, Burkina Faso, Nicaragua, India, and Turkey where viral and bacterial gastroenteritis prevalence of 70%, 64%, 61.1%, 60%, and 59.2% were reported respectively” Did the referenced studies use the same detection method? Here the authors do not specify the proportion of viral and bacterial detection singularly.

Response: This section is for the combine prevalence and there is another section where the proportion of viral and bacterial detection were specified singly. 

“Rotavirus infection was observed in this study to have the highest prevalence among the different microbial etiology of childhood diarrhoea screened for”: This sentence seems incomplete.

“This might not be unconnected with the fact that majority of children are not vaccinated against rotavirus infection and the environmental factors in which they live aids the spread of the virus”: As Nigeria vaccine was launched in August 2022, the verb mode should be reviewed. How do the authors conclude that the environmental factors would spread the virus? Which tests were performed to test this conclusion?

“In this study, a slight reduction in the prevalence of rotavirus gastroenteritis was observed (45.1%) compared to the report by Japhet et al. (2018) (47.6%) showing a 2.5% reduction: The authors describe reduction in rotavirus prevalence, which are the comparative data from the country?

Response

This has been expunged from the discussion in order to focus on our results as advised by the reviewer

“Till date, rotavirus vaccine is yet to be introduced into the free Nigeria national immunization program but is still being paid for by parents who can afford it. Low prevalence of RVA has been reported by countries where RVA vaccine is fully introduced into their national immunization scheme. As suggested by Japhet et al. (2018), Nigeria needs to consider including rotavirus vaccine into the free immunization given to children.”: Please consider to review the verb mode, as vaccine was introduced in August 2022.

Response: Thank you very much for the information on the current happenings about immunization of the rotavirus vaccine. However, we have made effort to reach out to all the healthcare centers where the participants were recruited for this study, and to date, they are yet to include free rotavirus vaccine in their scheme. The vaccine is only made available to those who can afford payment.

---

## [Decision Letter · Decision Letter 1]

26 Jan 2023

PONE-D-22-22211R1Rotavirus and bacterial diarrhoea among children in Ile-Ife, Nigeria: burden, risk factors and seasonality.PLOS ONE

Dear Dr. Japhet,

Thank you for submitting your manuscript to PLOS ONE. After careful consideration, we feel that it has merit but does not fully meet PLOS ONE’s publication criteria as it currently stands. Therefore, we invite you to submit a revised version of the manuscript that addresses the points raised during the review process.

We look forward to receiving your revised manuscript.

Kind regards,

Tacilta Nhampossa

Academic Editor

PLOS ONE

Reviewers' comments:

Reviewer's Responses to Questions

**Comments to the Author**

1. If the authors have adequately addressed your comments raised in a previous round of review and you feel that this manuscript is now acceptable for publication, you may indicate that here to bypass the “Comments to the Author” section, enter your conflict of interest statement in the “Confidential to Editor” section, and submit your "Accept" recommendation.

Reviewer #1: All comments have been addressed

Reviewer #2: (No Response)

2. Is the manuscript technically sound, and do the data support the conclusions?

Reviewer #1: Yes

Reviewer #2: No

3. Has the statistical analysis been performed appropriately and rigorously? 

Reviewer #1: Yes

Reviewer #2: Yes

4. Have the authors made all data underlying the findings in their manuscript fully available?

Reviewer #1: Yes

Reviewer #2: Yes

5. Is the manuscript presented in an intelligible fashion and written in standard English?

Reviewer #1: Yes

Reviewer #2: No

6. Review Comments to the Author

Reviewer #1: (No Response)

Reviewer #2: Dear Editor

Please see attached the comments to the authors after a second round of revision. Overall, the manuscript still needs further improvement.

7. PLOS authors have the option to publish the peer review history of their article (what does this mean?). If published, this will include your full peer review and any attached files.

Reviewer #1: No

Reviewer #2: No

---

## [Author Response · Author response to Decision Letter 1]

3 Mar 2023

I have responded to all the comments.

---

## [Decision Letter · Decision Letter 2]

26 Jul 2023

PONE-D-22-22211R2Rotavirus and bacterial diarrhoea among children in Ile-Ife, Nigeria: burden, risk factors and seasonality.PLOS ONE

Dear Dr. Japhet,

Thank you for submitting your manuscript to PLOS ONE. After careful consideration, we feel that it has merit but does not fully meet PLOS ONE’s publication criteria as it currently stands. Therefore, we invite you to submit a revised version of the manuscript that addresses the points raised during the review process.

We look forward to receiving your revised manuscript.

Kind regards,

Tacilta Nhampossa

Academic Editor

PLOS ONE

Journal Requirements:

Additional Editor Comments:

Please respond to all reviewer 2 comments

Reviewers' comments:

Reviewer's Responses to Questions

**Comments to the Author**

1. If the authors have adequately addressed your comments raised in a previous round of review and you feel that this manuscript is now acceptable for publication, you may indicate that here to bypass the “Comments to the Author” section, enter your conflict of interest statement in the “Confidential to Editor” section, and submit your "Accept" recommendation.

Reviewer #2: All comments have been addressed

2. Is the manuscript technically sound, and do the data support the conclusions?

Reviewer #2: Yes

3. Has the statistical analysis been performed appropriately and rigorously? 

Reviewer #2: Yes

4. Have the authors made all data underlying the findings in their manuscript fully available?

Reviewer #2: Yes

5. Is the manuscript presented in an intelligible fashion and written in standard English?

Reviewer #2: Yes

6. Review Comments to the Author

Reviewer #2: Date: 02 April 2023

To: PLOS ONE

Subject: Review of the manuscript – PONE-D-22-22211R1

Dear Academic Editor, PLOS ONE

Thank you for your invitation to review the third version of the manuscript “Rotavirus and bacterial diarrhoea among children in Ile-Ife, Nigeria: burden, clinical/environmental factors and seasonality”. The manuscript describes a basic hospital investigational of diarrhoea. The authors have improved several issues that were pointed, however, there is still need to correct the abstract. Please see my comments attached.

7. PLOS authors have the option to publish the peer review history of their article (what does this mean?). If published, this will include your full peer review and any attached files.

Reviewer #2: No

---

## [Author Response · Author response to Decision Letter 2]

3 Aug 2023

Academic Editor’s comment

If applicable, we recommend that you deposit your laboratory protocols in protocols.io to enhance the reproducibility of your results.

Response: The recommendation to deposit our protocol is not applicable to this study.

Response: We have carefully reviewed all our references, all the references cited in our manuscript are still available online, none is retracted.

Journal Requirements: Please upload your figure files to the Preflight Analysis and Conversion Engine (PACE) digital diagnostic tool, https://pacev2.apexcovantage.com/. PACE helps ensure that figures meet PLOS requirements.

Response: All our figures have been uploaded to the preflight analysis and conversion engine (PACE) as requested.

Reviewer’s comment

Again, we appreciate the invaluable suggestions of our indefatigable reviewer towards making our manuscript to be at its best. The suggestions on the abstract have been accepted and addressed below with slight modification to avoid repeated use of the word “usage” in a single paragraph.

1. Abstract

Background: Suggestion: “Diarrhea is a leading cause of death among under-five children globally, with sub-Saharan Africa alone accounting for 1/3 episodes yearly. Viruses, bacteria and parasites may cause diarrhea. Rotavirus is the most common viral etiology of diarrhoea in children less than five years globally. In Nigeria, there is scarce data on the prevalence/importance, burden, clinical/risk factors and seasonality of rotavirus and bacteria and this study aims to determine the role of rotavirus and bacteria on diarrheal cases in children less than five years in Ile-Ife, Nigeria. 

Response: This has been accepted and effected. 

Methods: Consider removing ..from 104 under five, because seems to be a result

Response: This has been removed. Thank you.

Conclusion: Suggestion: In this study, rotavirus was more prevalent than bacteria and occurred only in the dry season. Among bacterial etiologies, E. coli. was the most common detected. Differences in seasonal peaks of rotavirus and E.coli could be employed in diarrhoea management in Nigeria and other tropical countries for optimal usage of limited resources usage in preventing diarrhoea transmission and reducing indiscriminate antibiotics usage. 

Response: To avoid repeated use of the word “usage” which appeared thrice in the reviewer’s suggestion in a single paragraph, we reconstructed the conclusion thus:

In this study, rotavirus was more prevalent than bacteria and occurred only in the dry season. Among bacteria aetiologies, DEC was the most common detected. Differences in seasonal peaks of rotavirus and DEC could be employed in diarrhoea management in Nigeria and other tropical countries to ensure optimal usage of limited resources in preventing diarrhoea transmission and reducing indiscrimate use of antibiotics.

Notice that the word indiscriminate is wrongly written in your conclusion. E. coli must be replaced by DEC.

Thank you for your observation, we have now correctly spelt ‘indiscriminate’ and E.coli has been replaced with DEC.

---

## [Decision Letter · Decision Letter 3]

23 Aug 2023

Rotavirus and bacterial diarrhoea among children in Ile-Ife, Nigeria: burden, risk factors and seasonality.

PONE-D-22-22211R3

Dear Dr. Japhet,

We’re pleased to inform you that your manuscript has been judged scientifically suitable for publication and will be formally accepted for publication once it meets all outstanding technical requirements.

Kind regards,

Tacilta Nhampossa

Academic Editor

PLOS ONE

Additional Editor Comments (optional):

Reviewers' comments:

Reviewer's Responses to Questions

**Comments to the Author**

1. If the authors have adequately addressed your comments raised in a previous round of review and you feel that this manuscript is now acceptable for publication, you may indicate that here to bypass the “Comments to the Author” section, enter your conflict of interest statement in the “Confidential to Editor” section, and submit your "Accept" recommendation.

Reviewer #2: All comments have been addressed

2. Is the manuscript technically sound, and do the data support the conclusions?

Reviewer #2: Yes

3. Has the statistical analysis been performed appropriately and rigorously? 

Reviewer #2: Yes

4. Have the authors made all data underlying the findings in their manuscript fully available?

Reviewer #2: Yes

5. Is the manuscript presented in an intelligible fashion and written in standard English?

Reviewer #2: Yes

6. Review Comments to the Author

Reviewer #2: (No Response)

7. PLOS authors have the option to publish the peer review history of their article (what does this mean?). If published, this will include your full peer review and any attached files.

Reviewer #2: No

---

## [Editor Report · Acceptance letter]

4 Sep 2023

PONE-D-22-22211R3 

ROTAVIRUS AND BACTERIAL DIARRHOEA AMONG CHILDREN IN ILE-IFE, NIGERIA: BURDEN, RISK FACTORS AND SEASONALITY. 

Dear Dr. Japhet:

I'm pleased to inform you that your manuscript has been deemed suitable for publication in PLOS ONE. Congratulations! Your manuscript is now with our production department. 

Kind regards, 

on behalf of

Dr. Tacilta Nhampossa 

Academic Editor

PLOS ONE